# An open-source computational and data resource to analyze digital maps of immunopeptidomes

Etienne Caron[1]*, Lucia Espona[1], Daniel J Kowalewski[2,3], Heiko Schuster[2,3], Nicola Ternette[4], Adán Alpízar[5], Ralf B Schittenhelm[6], Sri H Ramarathinam[6], Cecilia S Lindestam Arlehamn[7], Ching Chiek Koh[1], Ludovic C Gillet[1], Armin Rabsteyn[2,3], Pedro Navarro[8], Sangtae Kim[9], Henry Lam[10], Theo Sturm[1], Miguel Marcilla[5], Alessandro Sette[7], David S Campbell[11], Eric W Deutsch[11], Robert L Moritz[11], Anthony W Purcell[6], Hans-Georg Rammensee[2,3], Stefan Stevanovic[2,3], Ruedi Aebersold[1,12]*

[1]Department of Biology, Institute of Molecular Systems Biology, ETH Zürich, Zurich, Switzerland; [2]Department of Immunology, Interfaculty Institute for Cell Biology, University of Tübingen, Tübingen, Germany; [3]DKFZ partner site Tübingen, German Cancer Consortium, Tübingen, Germany; [4]Target Discovery Institute Mass Spectrometry Laboratory, University of Oxford, Oxford, United Kingdom; [5]Proteomics Unit, Spanish National Biotechnology Centre, Madrid, Spain; [6]Department of Biochemistry and Molecular Biology, Monash University, Clayton, Australia; [7]La Jolla Institute for Allergy and Immunology, La Jolla, United States; [8]Institute for Immunology, University Medical Center of the Johannes Gutenberg University Mainz, Mainz, Germany; [9]Pacific Northwest National Laboratory, Richland, United States; [10]Division of Biomedical Engineering and Department of Chemical and Biomolecular Engineering, Hong Kong University of Science and Technology, Hong Kong, China; [11]Institute for Systems Biology, Seattle, United States; [12]Faculty of Science, University of Zurich, Zurich, Switzerland

*For correspondence: caron@imsb.biol.ethz.ch (EC); aebersold@imsb.biol.ethz.ch (RA)

Competing interests: The authors declare that no competing interests exist.

**Abstract** We present a novel mass spectrometry-based high-throughput workflow and an open-source computational and data resource to reproducibly identify and quantify HLA-associated peptides. Collectively, the resources support the generation of HLA allele-specific peptide assay libraries consisting of consensus fragment ion spectra, and the analysis of quantitative digital maps of HLA peptidomes generated from a range of biological sources by SWATH mass spectrometry (MS). This study represents the first community-based effort to develop a robust platform for the reproducible and quantitative measurement of the entire repertoire of peptides presented by HLA molecules, an essential step towards the design of efficient immunotherapies.

## Introduction

Next-generation immune-based therapies are expected to facilitate the eradication of intractable pathogens, cancer and autoimmune diseases (*Koff et al., 2013*). T cells play a critical role in such therapies by their ability to detect the presence of disease-specific antigens/peptides presented by major histocompatibility complex (MHC) molecules (human leukocyte antigen [HLA] molecules in humans). Under steady-state or pathological conditions, thousands of HLA class I-associated peptides of 8–12 amino acids in length are displayed on the surface of virtually all nucleated cells for scrutiny by

**eLife digest** The cells of the immune system protect us by recognizing telltale molecules produced by damaged and diseased cells, or by infection-causing microorganisms (which are also called pathogens). To help with this process, the cells in our bodies display small fragments of proteins (called peptides) on their surface that are then checked by the immune cells. Collectively, these peptides are referred to as the 'immunopeptidome', and deciphering the complexity of the human immunopeptidome is important for both basic research and medical science. Such an achievement would help to guide the development of next-generation vaccines and therapies against autoimmune disorders, infectious diseases and cancers.

In the past, immune peptides were mostly identified using a technique that is commonly called 'shotgun' mass spectrometry. However, this approach doesn't always provide reproducible results. In 2012, researchers reported the development of a new approach—which they called 'SWATH' mass spectrometry—that could yield more reproducible data.

Now, Caron et al.—including many of the researchers involved in the 2012 study—have developed a large collection of standardized tests that use SWATH mass spectrometry to analyze the human immunopeptidome. The workflow and the computational and data resources developed as part of this international effort are the first steps toward highly reproducible and measurable analyses of the immunopeptidome across many samples. Moreover, the large repository of assays generated by the project has been made public and will serve a large community of researchers, which should enable better collaborations.

In the future, SWATH mass spectrometry could be used as a robust technology for the reproducible detection and measurement of pathogen-specific or cancer-specific immune peptides. This could greatly help in the design of personalized immune-based therapies.

CD8+ T cells. HLA class II-associated peptides are 10–25 amino acids in length and are normally found on the surface of specialized antigen-presenting cells including macrophages and dendritic cells for presentation to CD4+ T cells. Collectively, HLA class I and class II peptides are referred to as the immunopeptidome, also known as HLA ligandome/peptidome (*Caron et al., 2011*; *Kowalewski et al., 2014*). The composition of the immunopeptidome in the human population is complicated by the presence of more than 3000 HLA alleles, resulting in a high diversity of peptide repertoires characterized by the presence of HLA allele-specific binding motifs (*Falk et al., 1991*). To be successful in designing efficient immunotherapies against autoimmunity, cancer and infectious diseases, it is becoming increasingly important to comprehensively map the complexity of the human immunopeptidome and to gain a more quantitative understanding of its dynamics in various disease states.

Mass spectrometry (MS) has evolved as the method of choice for the exploration of the human immunopeptidome (*Hunt et al., 1992*; *Admon and Bassani-Sternberg, 2011*; *Granados et al., 2015*). The largest HLA peptidomes reported to date using MS contain more than 10,000 class I or class II peptides (*Hassan et al., 2013*; *Bergseng et al., 2014*; *Bassani-Sternberg et al., 2015*). Estimates from various analytical and cell-based techniques also indicate that individual peptides are expressed on average at 50 copies per cell with extremes ranging from 1 to 10,000 copies per cell (*Granados et al., 2015*). Until recently, the most common strategy for the analysis of immunopeptidomes by MS has focused on the isolation of HLA-bound peptides by immunoaffinity chromatography and the collection of fragment ion spectra of selected peptides through automated MS operated in data-dependent acquisition (DDA) mode. Although DDA is a powerful strategy for exploring the peptidomic content of various cell and tissue types, it is not a reliable platform for solving problems that require the comparison of comprehensive, quantitative, and reproducible data sets across many samples or conditions. In fact, analyses of complex/unfractionated digests of cell lysate using DDA have shown that as many as 84% of peptides may remain unselected for fragmentation even though they are clearly detectable by the mass spectrometer (*Michalski et al., 2011*). Although the complexity of isolated HLA peptides is hardly comparable with that of cell lysate digests, as many as 20% of the selected HLA peptides can vary between replicate analyses of the same sample (*Granados et al., 2014*) (*Figure 1—figure supplement 1A*). A second strategy, referred

to as selected/multiple reaction monitoring (S/MRM), is a targeting MS technique capable of generating highly reproducible, quantitatively accurate and sensitive datasets (*Picotti and Aebersold, 2012*). S/MRM is, however, limited by its capacity to detect only tens to hundreds of peptides per sample injection and thus is not ideally suited to comprehensively quantify HLA peptidomes. To overcome this limitation, we recently introduced SWATH-MS, a new mass spectrometric technique that combines data-independent acquisition (DIA) with a targeted data extraction strategy (*Gillet et al., 2012*; *Röst et al., 2014*). In DIA mode, all peptides in a sample are fragmented and the corresponding fragment ion spectra are acquired, resulting in a digital recording of the peptide sample. DIA is an unbiased MS technique and therefore represents a suitable strategy for efficiently generating consistent, reproducible and quantitatively accurate measurements of peptides across multiple samples (*Gillet et al., 2012*; *Collins et al., 2013*; *Rosenberger et al., 2014*; *Röst et al., 2014*; *Guo et al., 2015*; *Liu et al., 2015*; *Schubert et al., 2015a*).

To extract quantitative information from digital SWATH-MS data, high-quality assay libraries are required. Such libraries contain retention-time and fragmentation information of the peptides to be targeted. Assay libraries are generated from native and/or synthetic peptides using a SWATH compatible mass spectrometer operated in DDA mode. To date, several generic SWATH assay libraries were generated for the analysis of proteomes in various species. These include *Mycobacterium tuberculosis* (*Schubert et al., 2015a*), *Saccharomyces cerevisiae* (*Selevsek et al., 2015*), and *Homo sapiens* (*Rosenberger et al., 2014*). Assay libraries were successfully employed to measure a limited number of MHC class I peptides by S/MRM in various contexts—that is, viral infection (*Croft et al., 2013*), autoimmunity (*Schittenhelm et al., 2014a*) and cancer (*Gubin et al., 2014*)—but have never been created for robust quantitative and high-throughput measurement of HLA-associated peptides by SWATH-MS.

For the SWATH-MS technology to meet its potential to support rapid advances in the design of next-generation vaccines and immunotherapies, comprehensive HLA peptide assay libraries have to be created and made readily available to basic and translational scientists. Generating such assay libraries could ultimately enable the fast and reproducible quantification of the entire repertoire of HLA peptides across many samples. Towards this end, we developed a workflow to (1) generate a pilot repository of HLA allele-specific peptide spectral and assay libraries, and to (2) analyze SWATH-MS HLA peptidomic data acquired from multiple international laboratories (*Figure 1*). In this study, libraries were created from natural and/or synthetic HLA class I and II peptides whereas analysis of SWATH-MS HLA peptidomic data focused mainly on naturally presented class I peptides.

## Results and discussion

Large-scale DDA-based identification of immunoaffinity purified HLA class I peptides is supported by several software tools (e.g., MaxQuant, Perseus or X-PRESIDENT) and results in thousands of unclassified peptides of various lengths. Since large HLA peptidomic datasets are generated at an increasing pace, additional computational frameworks facilitating the HLA annotation and storage of such datasets need to be developed. Here, we first created a computational workflow to support the identification, classification/annotation, visualization and storage of HLA peptidomic data in an allele-dependent manner. The software tools described in the section below enable (1) systematic annotation of peptides to their respective HLA allele, (2) visualization of HLA peptidomic datasets, and (3) generation of HLA class I allele-specific peptide spectral libraries, which can be converted into high quality assay libraries for the processing of SWATH-data (*Figure 2*, *Figure 2—figure supplement 1*, *Figure 2—source data 2* and *Supplementary file 1*).

To test our workflow, the generated data and computational resources, we first assessed the feasibility of generating HLA class I allele-specific peptide spectral libraries from a panel of fourteen PBMC samples (PBMC #1–14) expressing different combinations of HLA class I alleles. HLA class I-bound peptides were isolated from HLA-typed PBMC's by immunoaffinity chromatography and analyzed by DDA on an Orbitrap-XL mass spectrometer (*Figure 2* and *Figure 2—source data 1*). Peptides were identified using multiple open-source database search engines. The search identifications were combined and statistically scored using PeptideProphet and iProphet within the Trans-Proteomic Pipeline (TPP) as shown previously (*Figure 1*) (*Shteynberg et al., 2011*, *2013*). We next annotated the identified peptides to their respective HLA allele. Previously, HLA binding prediction algorithms such as SYFPETHI, NetMHC and SMM were used for manual or semi-automated

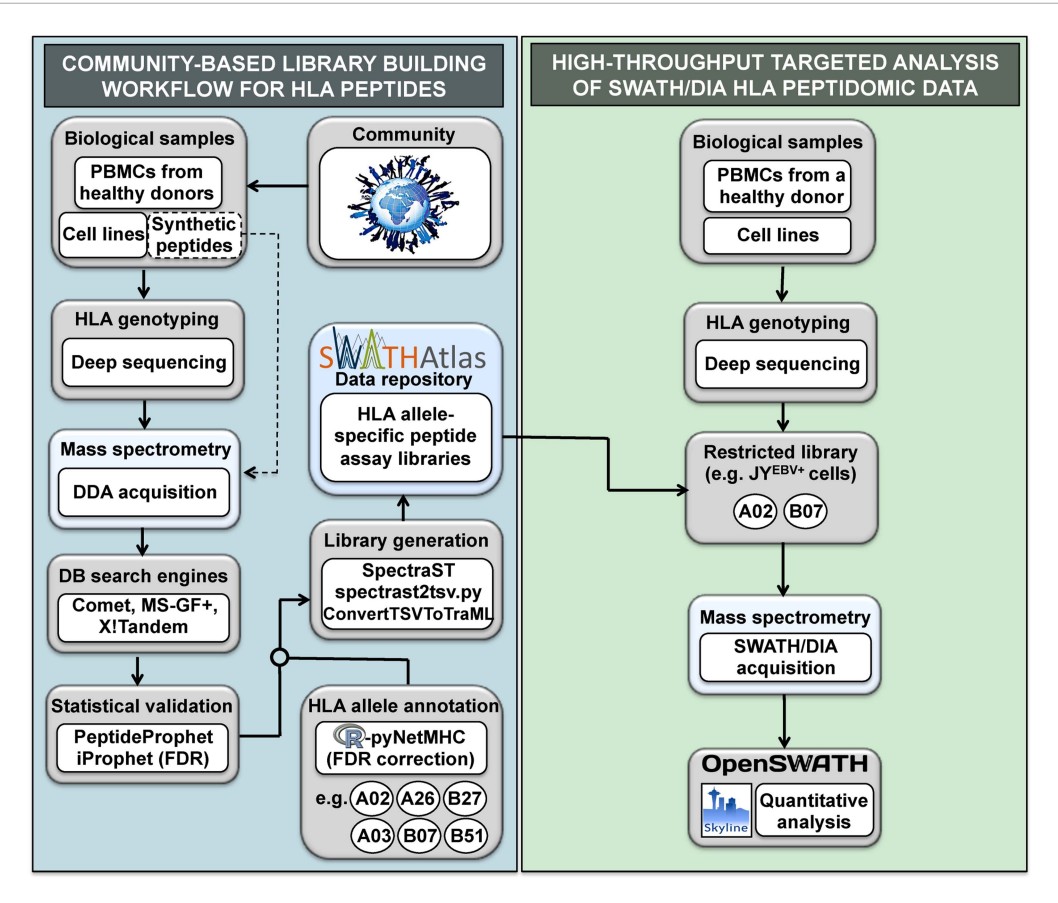

**Figure 1**. General workflow for building HLA allele-specific peptide assay libraries and for analyzing SWATH-MS HLA peptidomic data. (Left panel) A community-based repository of HLA class I allele-specific peptide spectral and assay libraries was created and stored in the SWATHAtlas database. HLA typed-biological samples and synthetic HLA peptides were used to build the repository. Our workflow integrates (1) data-dependent acquisition (DDA) of HLA peptidomic data, (2) multiple open-source database search engines and statistical validation tools, (3) HLA allele annotation of the identified peptides, and (4) spectral and assay library generation tools. (Right panel) HLA peptidomic data from HLA-typed biological samples were acquired in data-independent acquisition (DIA) mode. The matching HLA class I allele-specific peptide assay libraries were combined and DIA data were analyzed using the OpenSWATH and the Skyline software.

The following source data and figure supplements are available for figure 1:

**Source data 1**. Comparative analysis of DDA and SWATH-MS for the identification of HLA class I peptides.

**Figure supplement 1**. Reproducibility of DDA and SWATH-MS for the identification of HLA class I peptides.

**Figure supplement 2**. Combining results of three open-source database search engines in immunopeptidomics using iProphet.

**Figure supplement 3**. Combining both open-source and commercial database search engines in immunopeptidomics.

annotation of HLA peptides (*Fortier et al., 2008*; *Berlin et al., 2014*; *Granados et al., 2014*). Here, we designed a fully automated annotation strategy integrating the stand-alone software package of the HLA binding prediction algorithm NetMHC 3.4 with a set of in-house software tools (*Figure 2—figure supplement 1*). The in-house software tools enable an automated, consistent and effective annotation of the majority of the identified peptides to their respective HLA allele

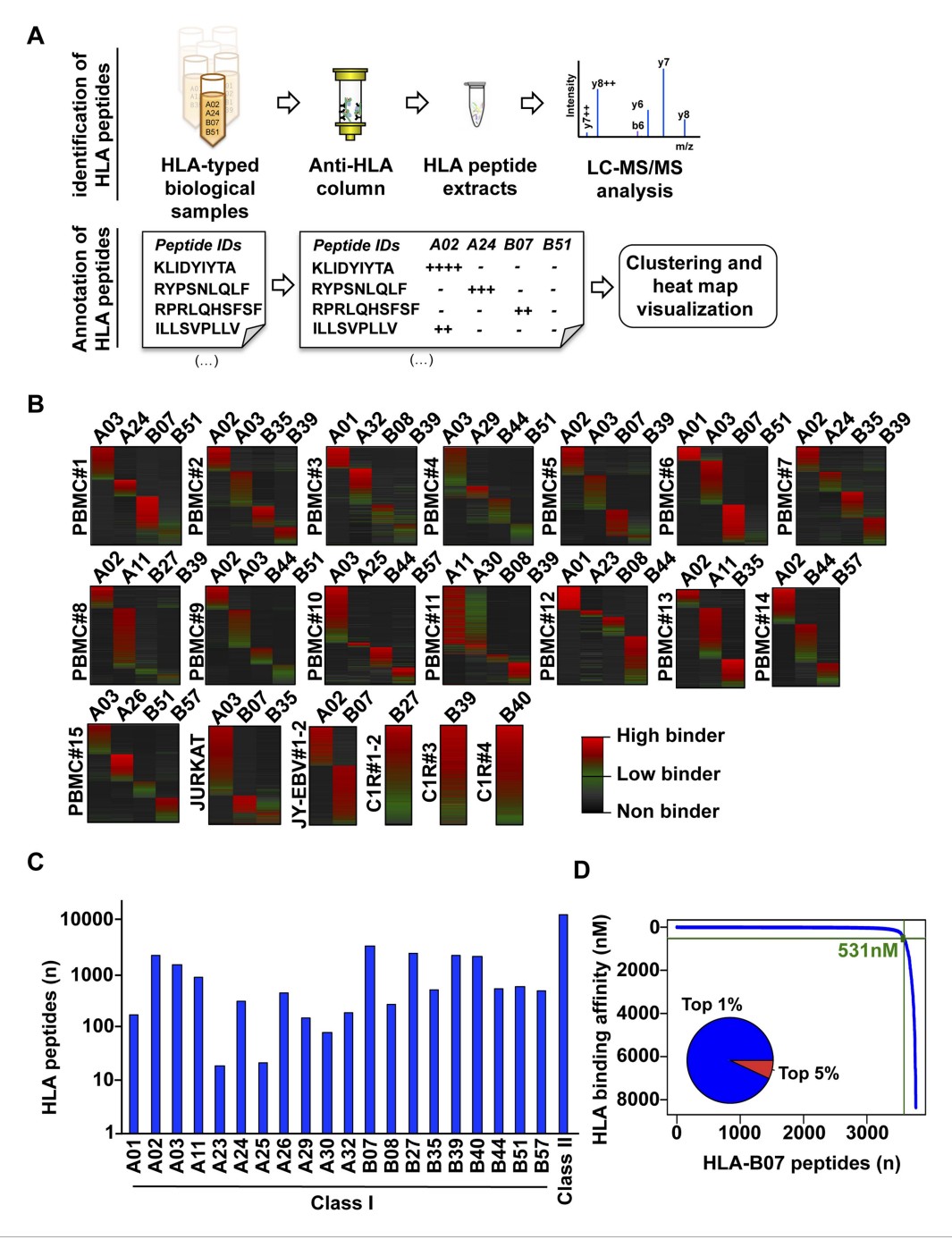

**Figure 2**. Content and analysis of the pilot repository. (**A**) HLA peptides were isolated by immunoaffinity chromatography and were annotated to their respective HLA alleles following DDA mass spectrometry. (**B**) Heat map visualization of HLA class I peptides identified from 20 HLA-typed biological samples. HLA-A and -B alleles are indicated for each sample. (**C**) 35,812 distinct class I and class II HLA peptides were identified, annotated, and used to build 32 and 11 HLA allele-specific peptide spectral and SWATH assay libraries, respectively. (**D**) The distribution curve shows that 95% of the HLA-B07-annotated peptides were predicted to bind the HLA molecule with an IC50 below 531 nM. Inner pie chart: we assessed the predicted HLA binding affinity of all peptides contained in individual source proteins. The pie chart shows that 92% of naturally presented HLA-B07 peptides were ranked in the top 1% (blue) of predicted peptides (see also *Figure 2—figure supplement 6*).

*Figure 2. continued on next page*

*Figure 2. Continued*

The following source data and figure supplements are available for figure 2:

**Source data 1**. Sources of HLA peptides used in this study.

**Source data 2**. Annotation of HLA peptides.

**Source data 3**. List of eluted HLA class I peptides that were identified at 1% and 5% peptide-level FDR.

**Source data 4**. HLA class I allele-specific peptide spectral libraries stored in PeptideAtlas.

**Source data 5**. HLA class I and II allele-specific peptide assay libraries stored in the SWATHAtlas database.

**Figure supplement 1**. Automated NetMHC-based method for annotating and visualizing HLA allele-specific peptides.

**Figure supplement 2**. Identification of HLA class I allele-specific peptides by DDA.

**Figure supplement 3**. Generation of assay libraries from a large collection of synthetic HLA class II peptides.

**Figure supplement 4**. Distribution curves of peptide binding affinities for different HLA-A and -B alleles (1% peptide-level FDR; 0.5% cFDR).

**Figure supplement 5**. Distribution curves of peptide binding affinities for different HLA-A and -B alleles (5% peptide-level FDR; 2.5% cFDR).

**Figure supplement 6**. Binding scores of naturally presented HLA-A and -B peptides contained in individual source proteins.

(*Supplementary file 1*). Briefly, each identified peptide was given a predicted HLA binding affinity ($IC_{50}$) for each of the HLA alleles expressed in the corresponding healthy donor. An HLA annotation score was then computed for each individual peptide by dividing its second best $IC_{50}$ value (i.e., the second best predicted allele) by its best $IC_{50}$ value (i.e., the best predicted allele). The higher this annotation score was, the higher the probability was for the peptide to be correctly annotated to a specific HLA allele. As an example, in PMBC#2, an annotation score of 77 was computed for the KLEEQARAK peptide by dividing 21,400 nM (second best $IC_{50}$ value predicted for HLA-B39) by 278 nM (best $IC_{50}$ value predicted for HLA-A03) (*Figure 2—figure supplement 1A*). Peptides with an HLA annotation score ≥3 (selected cutoff value; see 'Materials and methods' and *Supplementary file 1*) were systematically annotated to the allele predicted to bind best (e.g., HLA-A03 for the KLEEQARAK peptide). Using this scoring strategy, ~80% of all identified 8–12-mers were annotated to a specific HLA-A or -B allele (*Figure 2—source data 2*). HLA-A and -B alleles were prioritized due to the high reliability of the NetMHC 3.4 predictor for a broad diversity of HLA-A and -B alleles as well as for their high expression levels (*Kim et al., 2014*; *Bassani-Sternberg et al., 2015*; *Trolle et al., 2015*). Peptides with an annotation score below 3 were considered as non-annotated in this study and were discarded for the process of building the HLA allele-specific peptide spectral libraries. Tables including scored peptides were then used to generate heat maps and visualize HLA-A and -B peptidomes of PBMC's as described (*Figure 2B* and *Supplementary file 1*). Of note, allele-supertype peptides (i.e., peptides predicted to strongly bind more than one allele with an $IC_{50}$ below 500 nM) were curated in the output files but were not visualized on the heat maps in this study. A corrected false discovery rate (cFDR) was estimated for each PBMC sample following removal of all non-annotated contaminant peptides (*Figure 1—figure supplement 2* and *Figure 1—figure supplement 3*), resulting in a total of 4153 (peptide-level FDR 1%; average cFDR 0.5%) or 7921 (peptide-level FDR 5%; average cFDR 2.5%) distinct annotated peptides distributed across eighteen HLA class I alleles (*Figure 2—figure supplement 2A* and *Figure 2—source data 3*). All annotated peptides identified from the 14 PBMC samples were then used in SpectraST (*Lam et al., 2008*) to build

the HLA class I allele-specific peptide spectral libraries ('Materials and methods'). The same procedure was applied to peptides identified from JY[EBV+] and C1R cells. Notably, endogenous HLA-C04 peptides were recently shown to be significantly expressed on the surface of C1R cells (*Schittenhelm et al., 2014b*) and were therefore considered in this study. In total, 3528 HLA-A peptides, 4208 HLA-B peptides and 205 HLA-C04 peptides were recorded in the spectral libraries, which were then stored in the public PeptideAtlas database (*Figure 2—source data 4*). In summary, we generated a computational workflow to effectively annotate and visualize HLA peptidomic data, which were finally converted and stored into HLA allele-specific peptide spectral libraries consisting of consensus fragment ion spectra. This strategy could be further refined to collect, store and share HLA peptidomic information obtained from various cell lines and from larger cohorts of donors. Importantly, this computational approach can be broadly applied to generate SWATH-compatible assay libraries as described below.

Libraries of consensus fragment ion spectra can be converted into high quality assays for high-throughput targeted analysis of SWATH-MS data, an emerging approach for reproducible, consistent and accurate quantitative measurements of peptides (*Gillet et al., 2012*; *Collins et al., 2013*; *Rosenberger et al., 2014*; *Röst et al., 2014*; *Guo et al., 2015*; *Liu et al., 2015*; *Selevsek et al., 2015*; *Schubert et al., 2015a*). Here, we aimed at initiating a worldwide community-based effort to generate pilot HLA allele-specific peptide assay libraries that could be further used for the analysis of SWATH-MS HLA peptidomic data. Naturally presented and/or synthetic HLA class I and class II peptides were provided from six independent laboratories and were analyzed using four distinct TripleTOF 5600 MS instruments operated in DDA acquisition mode in four different international institutions. Naturally presented HLA class I peptides from JY[EBV+] (HLA-A02 and -B07), PBMC (HLA-A03, -A26, -B51 and -B57), and Jurkat (HLA-A03, -B07 and -B35) cells were isolated by immunoaffinity chromatography (*Figure 2—source data 1*). Natural class I peptides from three C1R cell lines—stably expressing HLA-C04 as well as HLA-B27, -B39 or -B40 molecules—were also isolated using the same procedure. Synthetic EBV-derived peptides known to bind HLA-A02 or -B07 were also used to build the libraries (*Figure 2—source data 2*). All laboratories used the spiked-in landmark iRT peptides for retention time normalization (*Escher et al., 2012*). The DDA data generated by the different groups were shared and pipelined through the computational workflow described above, resulting in the identification of 7668 (peptide-level FDR 1%; average cFDR 0.5%) or 11,275 (peptide-level FDR 5%; average cFDR 2.5%) distinct HLA class I peptides distributed across eleven different HLA class I alleles (*Figure 2—figure supplement 2B* and *Figure 2—source data 3*). To properly assess the efficiency of generating HLA peptide assay libraries from synthetic peptides, a large collection of 20,176 synthetic HLA class II peptides was analyzed by DDA using different mass spectrometers and fragmentation methods (*Figure 2—figure supplement 3* and *Figure 2—source data 2*). Our results show that a total of 15,875 peptides (~79%) were identified (*Figure 2—source data 2*). A large collection of synthetic HLA class I peptides was not available but could be used in the future to extend the contents of the present class I libraries derived from native peptides. All identified peptides were used to build the HLA allele-specific peptide assay libraries ('Materials and methods'). To date, the pilot libraries contain a total of 223,735 transitions for 26,857 unique peptides and were stored by class and allele in the SWATHAtlas database (*Figure 2—source data 5* and http://www.swathatlas.org). By using the automated HLA peptide annotation method described above, we observed that similar binding affinities were predicted for HLA class I peptides identified at peptide-level FDR 1% and peptide-level FDR 5% (*Figure 2—figure supplement 4* and *Figure 2—figure supplement 5*), suggesting that a large fraction of true positives were excluded at peptide-level FDR 1%. Our data also show that 95% of the annotated class I peptides in this study were predicted to bind their respective HLA molecules with an $IC_{50}$ ranging from 72 nM (for HLA-A01) to 5682 nM (for HLA-B51) at peptide-level FDR 1% (*Figure 2—figure supplement 4*). Similar results were obtained at peptide-level FDR 5% (*Figure 2—figure supplement 5*). This result supports a recent study indicating that HLA class I alleles are associated with peptide-binding repertoires of different affinity (*Paul et al., 2013*). Altogether, we demonstrated the feasibility of collecting DDA data from multiple international laboratories to generate standardized HLA allele-specific peptide assay libraries. We anticipate this global effort as a first step towards the development of a standardized Pan-human HLA peptide assay library, which could be used to rapidly and reproducibly quantify the entire repertoire of peptides presented by HLA molecules using SWATH-MS.

SWATH-MS is emerging as a robust next-generation proteomics technique for efficiently generating reproducible, consistent and quantitatively accurate measurements of peptides across multiple samples (*Gillet et al., 2012*; *Collins et al., 2013*; *Rosenberger et al., 2014*; *Röst et al., 2014*; *Guo et al., 2015*; *Liu et al., 2015*; *Selevsek et al., 2015*; *Schubert et al., 2015a*). To promote the worldwide development of SWATH-based MS platforms towards robust quantitative measurements of HLA peptidomes, we assessed whether the HLA allele-specific assay libraries described above could be used to extract quantitative information from digital SWATH maps acquired by different laboratories. Importantly, four independent laboratories generated their own digital SWATH maps using TripleTOF 5600 MS operated in DIA acquisition mode. Naturally presented HLA class I peptides were isolated from the cell types mentioned above (i.e., JY$^{EBV+}$, Jurkat, PBMC and C1R). Precursors in the range of 400–1200 Th were divided into 32 SWATH windows of 25 Da (*Gillet et al., 2012*). All ionized peptide precursors in this mass range were fragmented, generating comprehensive and quantitative digital fragment ion maps. The HLA peptidome of JY$^{EBV+}$ cells was analyzed using the OpenSWATH (*Röst et al., 2014*) software tool and a combined assay library containing 22,206 transitions for 1507 HLA-A02 and 2194 HLA-B07 peptides—the two dominant HLA alleles expressed on these cells. At an estimated peptide-level FDR of 1% (m-score < 0.01), a total of 3150 unique HLA class I peptides were identified from the digital SWATH map (*Figure 3A,B,C*, *Figure 3—figure supplement 1A,B*, *Figure 3—figure supplement 7* and *Figure 3—source data 1*). Notably, assays generated from the synthetic EBV-related class I peptides enabled the identification of one EBV-derived HLA-A02 peptide (*Figure 3C*), thereby demonstrating that building high-quality assay libraries from synthetic class I peptides of pathogen origin could be useful for the identification of non-self HLA-bound peptides by SWATH-MS. To analyze self-HLA peptides isolated from PBMC (HLA-A03, -A26, -B51 and -B57), Jurkat (HLA-A03, B07 and -B35), C1R-B27 (HLA-B27) and C1R-B40 (HLA-B40) cells, the matching HLA class I allele-specific peptide assay libraries were combined accordingly using SpectraST and then processed in the OpenSWATH software. High-throughput targeted analysis from these four additional peptidomic datasets indicated that ∼81% of HLA class I peptides present in an assay library could be extracted from a quantitative digital SWATH map in a cell type-independent manner (peptide-level FDR 1%) (*Figure 3—figure supplement 1C*, *Figure 3—figure supplements 2–6* and *Figure 3—source data 1*). We next optimized the SWATH acquisition conditions according to the size distribution of HLA class I peptides. Most class I peptide precursors (∼98%) fall within the range of 400–700 Th and were divided in 30 SWATH windows of 10 Da width each. Using SWATH data generated from JY$^{EBV+}$ cells, we found that narrowing the size of the windows by 2.5-fold resulted in a ∼13% fold-increase in the identification of class I peptides (*Figure 3—figure supplement 1A*). The $R^2$ value for SWATH-MS quantification was 0.979 from two technical replicates (*Figure 3D*). In accordance with previous studies, we also observed that the dynamic range of peptides quantified in different cell types using SWATH-MS, based on their signal intensity, was about 3-4 orders of magnitude (*Figure 3E*) (*Hassan et al., 2013*; *Bassani-Sternberg et al., 2015*). Altogether, we demonstrate the feasibility of an international effort to build standardized HLA allele-specific peptide assay libraries, which were used to extract quantitative information from digital SWATH maps acquired in different sites. We therefore provide a proof of concept that acquisition of SWATH-MS HLA peptidomic data may enable robust analysis of the human immunopeptidome on a global scale.

To further establish the robustness of SWATH-MS for the measurement of HLA-associated peptides, we tested whether the JY$^{EBV+}$ HLA peptidome could be reproducibly detected across multiple MS injections. For this purpose, we prepared a sample of class I peptides by immunoaffinity purification from JY$^{EBV+}$ cells and we acquired three datasets in SWATH mode. The datasets were analyzed using OpenSWATH and a combined HLA-A02 and -B07 peptide assay library as described above. At an estimated peptide-level FDR of 1%, a total of 2933 unique HLA class I peptides were identified by SWATH-MS and 2832 peptides (97%) were found in all the SWATH analyses (*Figure 1—figure supplement 1B*, *Figure 1—source data 1*). We then conducted a comparative analysis by acquiring three additional datasets in DDA mode from the same sample of class I peptides using the same chromatographic conditions. In total, 3153 HLA-A and -B peptides were identified at 1% peptide-level FDR and 1261 peptides (40%) were found in all the DDA analyses (*Figure 1—figure supplement 1A*, *Figure 1—source data 1*). Thus, the SWATH method clearly outperformed the DDA approach for the reproducible identification of JY$^{EBV+}$ HLA class I peptides across several technical replicates. Overall, our results indicate that SWATH-MS has the capability of detecting large numbers of HLA peptides across multiple injections at a high degree of reproducibility. By providing

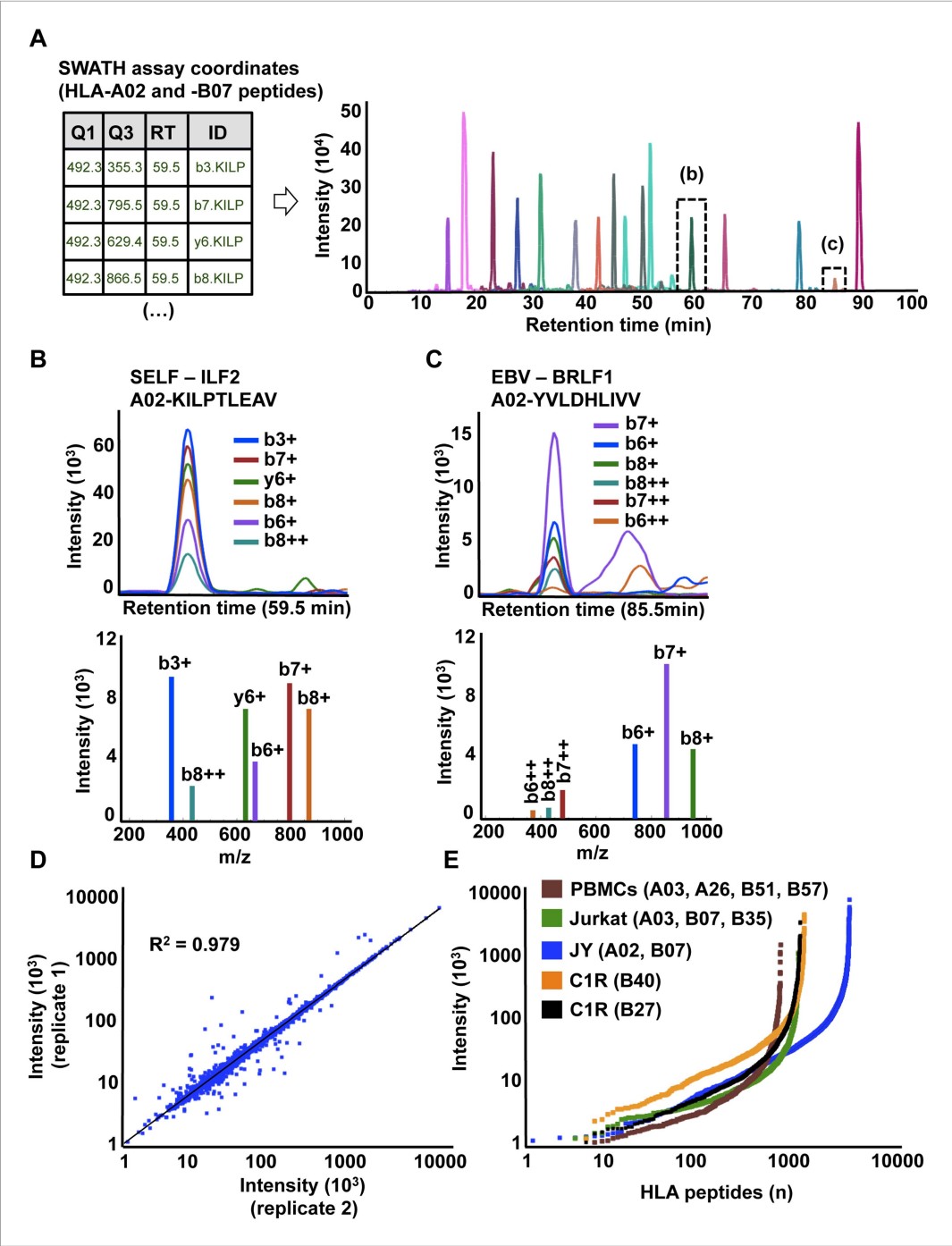

**Figure 3**. High-throughput targeted analysis of HLA peptidomic data by SWATH-MS. (**A**) SWATH-MS coordinates of two HLA class I allele-specific assay libraries (HLA-A02 and -B07) were combined to extract SWATH data generated from the HLA peptidome of JY[EBV+] cells. Sixteen summed transition groups are shown here for simplicity. (**B**, **C**) Visualization of two extracted SWATH transition groups corresponding to the self-HLA-A02 peptide, KILPTLEAV and the non-self HLA-A02 EBV peptide, YVLDHLIVV. (**D**) Reproducibility of intensity measurements for technical replicates. (**E**) Dynamic range of transition group intensities following targeted analysis of SWATH-MS HLA peptidomic data generated from various cell types expressing different combinations of HLA alleles. SWATH/DIA data were acquired in four independent international laboratories.

The following source data and figure supplements are available for figure 3:

**Source data 1**. OpenSWATH analysis.

*Figure 3. continued on next page*

*Figure 3. Continued*

**Figure supplement 1**. OpenSWATH analysis of HLA peptidomic data.

**Figure supplement 2**. OpenSWATH analysis and PyProphet statistics of HLA peptidomic data acquired at ETH Zurich, Switzerland.

**Figure supplement 3**. OpenSWATH analysis and PyProphet statistics of HLA peptidomic data acquired at ETH Zurich, Switzerland.

**Figure supplement 4**. OpenSWATH analysis and PyProphet statistics of HLA peptidomic data acquired at University of Oxford, UK.

**Figure supplement 5**. OpenSWATH analysis and PyProphet statistics of HLA peptidomic data acquired at Monash University, Australia.

**Figure supplement 6**. OpenSWATH analysis and PyProphet statistics of HLA peptidomic data acquired at Centro National de Biotechnologia, Madrid, Spain.

**Figure supplement 7**. Visualization and analysis of SWATH-MS HLA peptidomic data in Skyline.

a community resource for the continuous expansion of the library contents and by improving the performance of the OpenSWATH software, it can be expected that additional HLA peptides—including cryptic and mutant peptides—will be reproducibly identified and quantified from the same digital SWATH maps in the future.

The life sciences community greatly benefits from robust technologies such as microarrays and RNA-seq. Similarly, robust generation and analysis of quantitative digital maps of HLA peptidomes is expected to have important implications in basic and translational research as these will allow research groups to accurately investigate the dynamics of immunopeptidomes in various immune-related diseases such as autoimmunity, infectious diseases and cancers. For instance, reproducible digital mapping of tumor-specific mutant HLA peptides during cancer progression will facilitate stratification of patients who might best benefit from innovative immunotherapeutic interventions (*Gubin et al., 2014*; *Snyder et al., 2014*; *Schumacher et al., 2015*). The workflow and the computational and data resources presented in this community-based study is a first step towards highly reproducible and quantitative MS-based measurements of HLA peptidomes across many samples and could therefore be greatly beneficial in the design of personalized immune-based therapies. Moreover, the storage of HLA peptide spectral and assay libraries by class and allele in the SWATHAtlas database provides an initial framework to collect, organize and share HLA peptidomic data, thereby supporting the recently proposed Human Immunopeptidome and Vaccines Projects (*Admon and Bassani-Sternberg, 2011*; *Koff et al., 2014*).

## Materials and methods

### Blood samples, cell lines and synthetic peptides

PBMCs from healthy donors were isolated by density gradient centrifugation. Informed consent was obtained in accordance with the Declaration of Helsinki protocol. HLA typing was carried out by the Department of Hematology and Oncology, Tübingen, Germany. PBMCs were stored at −80°C until further use. JY[EBV+], Jurkat and C1R cells were cultured in RPMI supplemented with 10% fetal bovine serum, 50 IU/ml penicillin, and 50ug/ml streptomycin (Invitrogen, Life Technologies Europe BV, Zug, Switzerland). C1R cells were stably transfected with -B2705, -B3901 and -B4002 constructs, as described previously (*Marcilla et al., 2014*; *Schittenhelm et al., 2014a*). The EBV peptide was synthesized by Thermo Fischer Scientific (Ulm, Germany). The collection of 20,176 MTB peptides was synthesized by Mimotopes (Victoria, Australia) as described (*Lindestam Arlehamn et al., 2013*).

## Isolation of HLA peptides

HLA class I peptide complexes were isolated by standard immunoaffinity purification as described previously using the pan-HLA class I-specific mAb W6/32 (*Hunt et al., 1992*; *Croft et al., 2013*; *Kowalewski and Stevanovic, 2013*; *Marcilla et al., 2014*).

## RT normalization peptides

For the RT normalization and analysis, the peptides from the iRT Kit (Biognosys AG, Schlieren, Switzerland) were added to samples (see *Figure 2—source data 1*) prior to MS injection according to vendor instructions (*Escher et al., 2012*).

## DDA mass spectrometry

### AB SCIEX TripleTOF 5600+

Both naturally presented and synthetic HLA peptides were analyzed using a TripleTOF system (see *Figure 2—source data 1*) as described before (*Gillet et al., 2012*; *Röst et al., 2014*). Samples were analyzed on an Eksigent nanoLC (AS-2/1Dplus or AS-2/2Dplus) system coupled with a SWATH-MS-enabled AB SCIEX TripleTOF 5600+ System. The HPLC solvent system consisted of buffer A (2% acetonitrile and 0.1% formic acid in water) and buffer B (2% water with 0.1% formic acid in acetonitrile). The samples were separated in a 75 μm-diameter PicoTip emitter (New Objective, Woburn, MA) packed with 20 cm of Magic 3 μm, 200 Å C18 AQ material (Bischoff Chromatography, Leonberg, Germany). The loaded material was eluted from the column at a flow rate of 300 nl/min with the following gradient: linear 2–35% B over 120 min, linear 35–90% B for 1 min, isocratic 90% B for 4 min, linear 90–2% B for 1 min and isocratic 2% solvent B for 9 min. The mass spectrometer was operated in DDA top20 mode, with 500 and 150 ms acquisition time for the MS1 and MS2 scans respectively, and 20 s dynamic exclusion. Rolling collision energy with a collision energy spread of 15 eV was used for fragmentation.

### Thermo scientific orbitrap ELITE

Mtb synthetic peptides were analyzed on an Eksigent LC system coupled to an LTQ-Orbitrap ELITE mass spectrometer. Peptides were separated on a custom C18 reversed phase column (150 mm i.d. × 100 mm, Jupiter Proteo 4 mm, Phenomenex) using a flow rate of 600 nl min$^{-1}$ and a linear gradient of 3–60% aqueous ACN (0.2% formic acid) in 120 min. Full mass spectra were acquired with the Orbitrap analyser operated at a resolving power of 30,000 (at m/z 400). Mass calibration used an internal lock mass (protonated (Si(CH3)2O))6; m/z 445.120029) and mass accuracy of peptide measurements was within 5 p.p.m. MS/MS spectra were acquired in CID and HCD mode with a normalized collision energy of 35%. Up to ten precursor ions were accumulated to a target value of 50,000 with a maximum injection time of 300 ms and fragment ions were transferred to the Orbitrap analyser operating at a resolution of 15,000 at m/z 400.

### Thermo scientific orbitrap XL

Naturally presented HLA class I peptides from several PBMC samples (see *Figure 2—source data 1*) were also analyzed by reversed-phase liquid chromatography (nano-UHPLC, UltiMate 3000 RSLCnano; Thermo Fisher, Waltham, MA, USA) coupled with an LTQ Orbitrap XL hybrid mass spectrometer. Samples were analyzed in five technical replicates. Sample volumes of 5 μl (sample shares of 20%) were injected onto a 75 μm × 2 cm trapping column (Acclaim PepMap RSLC; Thermo Fisher) at 4 μl/min for 5.75 min. Peptide separation was subsequently performed at 50°C and a flow rate of 175 nl/min on a 50 μm × 50 cm separation column (Acclaim PepMap RSLC; Thermo Fisher) applying a gradient ranging from 2.4 to 32.0% of acetonitrile over the course of 140 min. Eluting peptides were ionized by nanospray ionization and analyzed in the mass spectrometer implementing a top five CID method generating fragment spectra for the five most abundant precursor ions in the survey scans. Resolution was set to 60,000. For HLA class I ligands, the mass range was limited to 400–650 m/z with charge states 2 and 3 permitted for fragmentation.

## Database search engines and statistical validation

All raw instrument data were centroided and processed as described previously (*Collins et al., 2013*; *Rosenberger et al., 2014*). The datasets were searched individually using X!tandem (*Craig et al., 2004*), MS-GF+ (*Kim and Pevzner, 2014*) and Comet (*Eng et al., 2012*) against the full non-redundant, canonical human genome as annotated by the UniProtKB/Swiss-Prot (2014_02) with

20,270 ORFs and appended iRT peptide and decoy sequence. Oxidation (M) was the only variable modification. Parent mass error was set to $\pm5$ p.p.m., fragment mass error was set to $\pm0.5$ Da. The search identifications were then combined and statistically scored using PeptideProphet (*Keller et al., 2002*) and iProphet (*Shteynberg et al., 2011*) within the TPP (4.7.0) (*Keller et al., 2005*). All peptides with an iProbability/iProphet score above 0.7 were exported in Excel. Assumed charges were also exported, as this information is needed in SpectraST. Length considered was 8–12 residues for class I HLA peptides. FDR was manually estimated based on the target-decoy approach (*Elias and Gygi, 2007*). Peptides (1% and 5% peptide-level FDR) were then exported to a .txt file for annotation to their respective HLA allele.

## HLA allele annotation

Annotation of the identified peptides (1% and 5% peptide-level FDR) to their respective HLA allele was performed automatically by integrating the stand-alone software package of NetMHC 3.4 (*Lundegaard et al., 2008*) with our in-house software tools (*Supplementary file 1* and *Source code 1*). An HLA annotation score was computed by the software tools for individual peptides (*Figure 2—figure supplement 1*). A predefined cutoff score of 3 was then used to annotate each peptide to their respective HLA allele. A cutoff value of 3 was selected because >90% of the identified peptides with an annotation score above 3 have a predicted $IC_{50}$ below 1000 nM. FDR was corrected from the list of annotated HLA peptides based on the target-decoy approach (*Elias and Gygi, 2007*). The software tools were used to process and visualize the peptidomic datasets. The final lists of HLA-allele specific peptides were exported into a .txt file and used in SpectraST for library generation.

## Generation of HLA allele-specific peptide spectral and assay libraries

This section was adapted from *Schubert et al. (2015b)*. The parameters below were used for Spectrast (*Lam et al., 2008*). Exact meaning of each parameter can be found in the following link: http://tools.proteomecenter.org/wiki/index.php?title=Software:SpectraST. Spectrast was used in library generation mode with CID-QTOF settings (-cICID-QTOF) for the Triple-TOF 5600$^+$ or CID (default) settings for the Orbitrap-XL and Orbitrap-ELITE. Retention times were normalized against the iRT Kit peptide sequences (-c_IRTiRT.txt -c_IRR). Only HLA-allele specific peptide ions were included for library generation (-cT):

spectrast -cNSpecLib_celltype_allele_fdr_iRT -cICID-QTOF -cTReference_celltype_allele_fdr.txt -cP0.7 -c_IRTiRT.txt -c_IRR iprophet.pep.xml

A consensus library was then generated:

spectrast -cNSpecLib_cons_celltype_allele_fdr_iRT -cICID-QTOF -cAC SpecLib_celltype_allele_fdr_iRT.splib

HLA-allele specific consensus libraries were merged:

spectrast -cNSpecLib_cons_celltype_alleles_fdr_iRT -cJU -cAC SpecLib_celltype_allele1_fdr_iRT.splib SpecLib_celltype_allele2_fdr_iRT.splib SpecLib_celltype_allele3_fdr_iRT.splib SpecLib_celltype_allele4_fdr_iRT.splib

The script spectrast2tsv.py (msproteomicstools 0.2.2; https://pypi.python.org/pypi/msproteomicstools) was then used to generate the HLA-allele specific peptide assay library with the following recommended settings:

spectrast2tsv.py -l 350,2000 -s b,y -x 1,2 -o 6 -n 6 -p 0.05 -d -e -w swaths.txt -k openswath -a SpecLib_cons_celltype_alleles_fdr_iRT_openswath.csv SpecLib_cons_celltype_alleles_fdr_iRT.sptxt

The _openswath.csv file was then converted into a .tsv file and opened in Excel. Reference coordinates for the 11 iRT peptides were confirmed and any remaining decoy sequences were removed. The file was then saved in .txt format and then converted back in .csv format. The OpenSWATH tool ConvertTSVToTraML converted the TSV/CSV file to TraML:

ConvertTSVToTraML -in SpecLib_cons_celltype_alleles_fdr_iRT_openswath.csv -out SpecLib_cons_celltype_alleles_fdr_iRT.TraML

Decoys were appended to the TraML assay library with the OpenSWATH tool OpenSwathDecoyGenerator as described before (*Rosenberger et al., 2014*; *Röst et al., 2014*; *Schubert et al., 2015b*) in reverse mode with a similarity threshold of 0.05 Da and an identity threshold of 1:

OpenSwathDecoyGenerator -in SpecLib_cons_celltype_alleles_fdr_iRT.TraML -out SpecLib_cons_celltype_alleles_fdr_iRT_decoy.TraML -method shuffle -append -exclude_similar

The library was then uploaded into the iPortal workflow for SWATH data analysis (see below).

## DIA mass spectrometry (SWATH-MS)

For SWATH-MS data acquisition, the same mass spectrometer and LC-MS/MS setup was operated essentially as described before (*Collins et al., 2013*; *Rosenberger et al., 2014*) using 32 windows of 25 Da effective isolation width (with an additional 1 Da overlap on the left side of the window) and with a dwell time of 100 ms to cover the mass range of 400–1200 m/z in 3.3 s. Before each cycle, an MS1 scan was acquired, and then the MS2 scan cycle started (400–425 m/z precursor isolation window for the first scan, 424–450 m/z for the second... 1,174–1200 m/z for the last scan). The collision energy for each window was set using the collision energy of a 2+ ion centered in the middle of the window with a spread of 15 eV. Four independent international laboratories acquired their own SWATH maps using the settings described above: (1) Antony Purcell, Monash University; (2) Nicola Ternette, University of Oxford; (3) Miguel Marcilla, Spanish National Biotechnology Center; (4) Ruedi Aebersold, ETH-Zurich.

## SWATH-MS data analysis

The iPortal workflow was used for data analyses (*Kunszt et al., 2014*). The OpenSWATH analysis workflow (OpenSWATHWorkflow) (http://www.openswath.org) was implemented in the iPortal workflow. The parameters were selected analogously to the ones described before (*Röst et al., 2014*): min_rsq: 0.95, min_coverage: 0.6, min_upper_edge_dist: 1, mz_extraction_window: 0.05, rt_extraction_window: 600, extra_rt_extraction_window: 100. pyprophet (https://pypi.python.org/pypi/pyprophet) was run on the OpenSwathWorkflow output adjusted to contain the previously described scores (xx_swath_prelim_score, bseries_score, elution_model_fit_score, intensity_score, isotope_correlation_score, isotope_overlap_score, library_corr, library_rmsd, log_sn_score, massdev_score, massdev_score_weighted, norm_rt_score, xcorr_coelution, xcorr_coelution_weighted, xcorr_shape, xcorr_shape_weighted. yseries_score) (*Röst et al., 2014*). Assay libraries were loaded into Skyline and SWATH traces were analyzed as described previously (*Schubert et al., 2015b*). Advanced protocols for analysis of SWATH/DIA data can be downloaded from the website: http://skyline.maccosslab.org.

## Acknowledgements

We thank Ben Collins, Yansheng Liu, Tatjana Sajic and Olga Schubert for instrument maintenance and for technical support. We thank Emanuel Schmid, Lorenz Blum, Hannes Röst, George Rosenberger and Ulrich Omasits for assistance with the computational analysis. We thank Valeria de Azcoitia for commenting this manuscript as well as all members of the Aebersold laboratory for discussions.

## Additional information

### Funding

| Funder | Grant reference | Author |
|---|---|---|
| National Health and Medical Research Council (NHMRC) | 1022509 and 1085017 | Anthony W Purcell |
| National Institutes of Health (NIH) | HHSN272201200010C and HHSN272200900044C | Cecilia S Lindestam Arlehamn, Alessandro Sette |
| European Research Council (ERC) | ERC-2008-AdG_20080422 | Ruedi Aebersold |
| Schweizerische Nationalfonds zur Förderung der Wissenschaftlichen Forschung | 3100A0-688 107679 | Ruedi Aebersold |
| European Commission (EC) | SysteMtb, 241587 | Ruedi Aebersold |
| German Cancer Consortium (DKTK) | | Daniel J Kowalewski, Heiko Schuster, Hans-Georg Rammensee, Stefan Stevanovic |

| Funder | Grant reference | Author |
|---|---|---|
| Bundesministerium für Bildung und Forschung | e:Bio Express2Present, 0316179C | Pedro Navarro |
| Forschungszentrum Immuntherapie (FZI) | of the Johannes Gutenberg University Mainz | Pedro Navarro |
| Ministerio de Economía y Competitividad | Carlos III Health Institute (ISCIII) (ProteoRed-PRB2, PT13/0001) | Miguel Marcilla |
| European Commission (EC) | Marie Curie Intra-European Fellowship | Etienne Caron |
| Schweizerische Nationalfonds zur Förderung der Wissenschaftlichen Forschung | Postdoc Mobility Fellowship | Ralf B Schittenhelm |
| National Institute of General Medical Sciences (NIGMS) | R01GM087221 and 2P50 GM076547/Center for Systems Biology | David S Campbell, Eric W Deutsch, Robert L Moritz |

The funders had no role in study design, data collection and interpretation, or the decision to submit the work for publication.

## Author contributions

EC, Conception and design, Acquisition of data, Analysis and interpretation of data, Drafting or revising the article; LE, CCK, LCG, PN, SK, HL, Analysis and interpretation of data, Drafting or revising the article; DJK, HS, Acquisition of data, Analysis and interpretation of data, Drafting or revising the article; NT, AA, RBS, SHR, MM, AWP, Acquisition of data, Drafting or revising the article; CSLA, AR, Acquisition of data, Drafting or revising the article, Contributed unpublished essential data or reagents; TS, AS, DSC, Analysis and interpretation of data, Drafting or revising the article, Contributed unpublished essential data or reagents; EWD, RLM, Conception and design, Drafting or revising the article, Contributed unpublished essential data or reagents; H-GR, SS, RA, Conception and design, Drafting or revising the article

## Author ORCIDs
Etienne Caron, http://orcid.org/0000-0003-2770-6970

## Ethics

Human subjects: Informed consent was obtained in accordance with the Declaration of Helsinki protocol. The study was performed according to the guidelines of the local ethics committee (University of Tubingen, Germany).

# Additional files

## Supplementary files

• Supplementary file 1. Description of the Python and the R scripts for the automated annotation and visualization of HLA peptidomic data.

• Source code 1. Python and R scripts.

## Major datasets

The following datasets were generated:

| Author(s) | Year | Dataset title | Dataset ID and/or URL | Database, license, and accessibility information |
|---|---|---|---|---|
| Caron et al, | 2015 | Mass spectrometry discovery peptidomics data (centroided mzXML and identified peptides in pepXML report) used to generate the HLA-allele specific peptide spectral and assay libraries | http://proteomecentral.proteomexchange.org | Publicly available at the ProteomeXchange (Accession no: PXD001872). |

| Author(s) | Year | Dataset title | Dataset ID and/or URL | Database, license, and accessibility information |
|---|---|---|---|---|
| Caron et al, | 2015 | HLA allele-specific peptide spectral libraries (SpectraST format) and assay libraries (CSV, TraML) available for different SWATH-MS data analysis tools | www.swathatlas.org | Publicly available at the SWATHAtlas. Additional allele-specific spectral libraries (without RT normalization) are available at the PeptideAtlas (www.peptideatlas.org) with the dataset identifier PASS00666. |
| Caron et al, | 2015 | Mass spectrometry SWATH-MS data (instrument raw/wiff files and identified peptides in OpenSWATH report) | http://proteomecentral.proteomexchange.org | Publicly available at the ProteomeXchange (Accession no: PXD001904). |

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
