## [Decision Letter]

Thank you for submitting your work entitled “An open-source computational and data resource to analyze digital maps of immunopeptidomes” for peer review at *eLife*. Your submission has been favorably evaluated by Tadatsugu Taniguchi (Senior editor), Arup K Chakraborty (Reviewing editor), and two reviewers.

The reviewers have discussed the reviews with one another and the Reviewing editor has drafted this decision to help you prepare a revised submission.

Summary:

This manuscript describes the use of the SWATH-MS methodology for identification and cataloging of HLA peptide repertoires – viz., 'the Immunopeptidome'. The manuscript is based on an international collaboration between a few laboratories specializing in HLA immunopeptidome analysis and the laboratory of Dr. Aebersold, who developed the SWATH_MS methodology. In the past, HLA peptides were mostly identified by shotgun proteomics, which is based on the selection of peptides from purified pools of peptides for fragmentation in the mass spectrometer during the LC-MS-MS analysis. The SWATH_MS approach does not select specific peptides for fragmentation, but instead scans repetitively incremental mass windows of about 25 mass units, every time fragmenting all the peptides that are present in each of these mass windows. The method allows fragmenting all the peptides which have the mass divided by charge of the mass region being scanned more than once. The advantage of this approach is the better reproducibility of the data while the disadvantage is the lower sensitivity and the need to establish beforehand a library of spectra of the individual peptides that will be used for identification by the SWATH-MS approach. The use of the peptide preparations and the LC-MS-MS data from the different collaborating labs resulted in successful establishment of a large repository of peptides and their spectra libraries in a format that will be made public and will serve a large community of researchers, enabling better collaborations.

Overall, the work reported in this paper could be a significant resource for the community. But, the following issues need to be addressed to further establish the robustness of the method.

Major points:

1) In the Introduction, it is suggested that Data-Dependent Acquisition (DDA) used for such analyses of MHC peptidomes results in less reproducible results. This is a well-known fact, but it would be good to show the reproducibility of the results of the same three raw data files when analyzed by the SWATH-MS method and a figure with parallel Venn diagrams indicating how many peptides are identified in each analysis in total, and how many are shared between the runs.

2) Peptide presentation by HLA C is ignored because it is claimed that they are expressed in low amounts. In Schittenhelm et al. (Tissue Antigens, 2014, 83, 174-179, 2014), an HLA immunopeptidome of HLA C is reported using the C1R cell line. Since the same cell line is used in the present study, ignoring peptide presentation by HLA C seems to be inappropriate. This point needs clarification or inclusion of HLA C in the analysis.

3) Several search engines (Comet, MS-GF+, X!Tandem) were used to identify peptide sequences from the mass spectrometry data. As can be seen from Figure 1—figure supplement 2, different engines identify quite different numbers of potential peptides. How should these results be interpreted? Should the union or intersection of these peptide sets be used?

4) The HLA annotation score, based on the predicted IC_50_ binding affinities from NetMHC server, is used to associate peptides with particular HLA alleles. In a similar server, Immune Epitope Database (IEDB), sometimes the values of experimentally measured and predicted values of IC_50_ can differ by a factor of 10 or more. Using the NetMHC server are the results more robust, thus allowing use of cutoff of value of 3 for the HLA annotation score?

Minor points:

1) In the first paragraph of Results and Discussion, the comment “no reference computational framework is currently available to facilitate the analysis of such datasets” is not entirely correct, since software tools, such as MaxQuant, Perseus, or X-PRESIDENT can handle HLA-peptidome data without effort (for example: see the second reference cited, [2]).

2) In the second paragraph of Results and Discussion: “of all identified peptides to their respective HLA allele” is an overstatement, since significant parts of the identified peptides are not annotated to their respective HLA allele.

3) In the third paragraph of Results and Discussion: “Three synthetic EBV-derived peptides were also used to build the HLA-A02 and -B07 library. How can three peptide by useful for building two libraries? Some clarification is required.

4) In the fourth paragraph of Results and Discussion: “Class I peptide precursors fall within the range of 400-700 Th” – this is not correct, since many peptides fall outside this range. We suggest clarifying the percentage of the peptides which fall within this range, and indicating if the loss of these peptides compensate for the additional 17% gained by use of this narrow range. Also, why was the 400-650 mass range selected?

5) In the subsection headed “Isolation of HLA peptides”: We suggest adding the reference by [20] to the references for the method of affinity purification and LC-MS-MS analysis of MHC peptidome.

6) In the subsection “Generation of HLA allele-specific peptide spectral and assay libraries”: Please clarify that the parameters are used for Spectrast, and explain what they mean for readers that do not use Spectrast.

---

## [Author Response]

*1) In the Introduction, it is suggested that Data-Dependent Acquisition (DDA) used for such analyses of MHC peptidomes results in less reproducible results. This is a well-known fact, but it would be good to show the reproducibility of the results of the same three raw data files when analyzed by the SWATH-MS method and a figure with parallel Venn diagrams indicating how many peptides are identified in each analysis in total, and how many are shared between the runs*.

This is an important point and we performed the analysis suggested by the reviewers. HLA class I peptides were freshly isolated from JY cells and six technical replicates were consecutively injected in a TripleTOF 5600 – three datasets were acquired in DDA mode and three datasets were acquired in SWATH/DIA mode. We inserted a new paragraph in the main text of the manuscript to describe the results of this analysis. We also added a new Venn diagram as suggested (Figure 1—figure supplement 1) as well as a new supplementary table ([Supplementary-material SD1-data]). In the new version of the manuscript, we now mention (Results and Discussion): “To further establish the robustness of SWATH-MS […] reproducibly identified and quantified from the same digital SWATH maps in the future.”

*2) Peptide presentation by HLA C is ignored because it is claimed that they are expressed in low amounts. In Schittenhelm et al. (*[41]*, 83, 174-179, 2014), an HLA immunopeptidome of HLA C is reported using the C1R cell line. Since the same cell line is used in the present study, ignoring peptide presentation by HLA C seems to be inappropriate. This point needs clarification or inclusion of HLA C in the analysis*.

We thank the reviewers for pointing this out. In this study, we focused on HLA-A and -B alleles because of the high reliability of the NetMHC 3.4 predictor for a wide diversity of HLA-A and -B alleles as well as for their high expression levels (24; 2; 49). We now mention (Results and Discussion): “HLA-A and -B alleles were prioritized due to the high reliability of the NetMHC 3.4 predictor for a broad diversity of HLA-A and -B alleles as well as for their high expression levels (24; 2; 49).”

Please note that a large number of HLA-C alleles will be fully integrated in the next version of the workflow. For instance, the next upgrade will integrate additional epitope prediction algorithms such as NetMHCpan and others from IEDB and will annotate supertype peptides. We also plan to integrate a predictor-independent strategy [e.g. alignment- and clustering-based approach similar to GibbsCluster and NNalign (Andreatta et al. 2013; Nielson et al. 2009)] since the HLA annotation score used in this manuscript can only be calculated for well-characterized HLA alleles. Nevertheless, it is correct that C1R cells express a significant number of HLA-C04 peptide ligands. Using C1R cells, Schittenhelm et al. have indeed identified 734 HLA-C04 specific peptides from 10 independent immunoaffinity purifications (Schittenhelm et al., 2014). In our study, 205 HLA-C04 peptide ligands expressed on the surface of C1R cells were identified and are now included in the new version of the manuscript. We now state: “Notably, endogenous HLA-C04 peptides were recently shown to be expressed on the surface of C1R cells (41) and were therefore considered in this study. In total, 3,528 HLA-A peptides, 4,208 HLA-B peptides and 205 HLA-C04 peptides were recorded in the spectral libraries”. We also state: “Natural class I peptides from three C1R cell lines – stably expressing HLA-C04 as well as HLA-B27, -B39 or -B40 molecules – were also isolated using the same procedure.”

We updated the total number of unique peptides, alleles and transitions in the main text (Results and Discussion) and an HLA-C04-specific peptide assay library is now available in the SWATH Atlas repository: https://db.systemsbiology.net/sbeams/cgi/PeptideAtlas/GetDIALibs?SBEAMSentrycode=HLASWATH2015. We also updated Figure 2—figure supplement 2 and [Supplementary-material SD3-data]. However, we did not modify the Figure 2 as the focus of this study is on HLA-A and -B alleles and because HLA-C04 peptide ligands would be poorly represented on the heat maps (see Figure 4).

Author response image 1.Heat map visualization of HLA-B27 and -C04 peptide ligands isolated from C1R cells.**DOI:**
http://dx.doi.org/10.7554/eLife.07661.033

*3) Several search engines (Comet, MS-GF+, X!Tandem) were used to identify peptide sequences from the mass spectrometry data. As can be seen from*
Figure 1—figure supplement 2*, different engines identify quite different numbers of potential peptides. How should these results be interpreted? Should the union or intersection of these peptide sets be used*?

Indeed, clarification is needed to explain what are the peptide sets that were used to build the spectral libraries. Combining results of multiple search engines in proteomics has been shown to be beneficial using particular softwares such as MSblender, PepArML and iProphet (47). In Figure 1—figure supplement 2, the venn diagrams were essentially used to roughly compare and visualize the numbers of HLA class I peptides identified by different search engines. However, neither the unions nor the intersections of the venn diagrams were used to build the spectral libraries. Here, the software iProphet was used to combine the search results generated by Comet, MS-GF+ and X!Tandem. In fact, iProphet allows accurate and effective integration of the results from multiple database search engines applied to the same data (46). In the revised version of the manuscript, we added a new table in Figure 1—figure supplement 2. The new table shows the numbers of HLA class I peptides obtained from the iProphet combined search results that were used to build the spectral libraries. The new table also shows the sum of peptides identified by the three search engines (Union) as well as the number of overlapping peptides (Intersection) for each Venn diagram/sample. Notably, the numbers of peptides obtained from ‘iProphet’ and ‘Union’ are very similar although not identical. In the revised version of the manuscript, we now state (Results and Discussion): “Peptides were identified using multiple open-source database search engines. The search identifications were combined and statistically scored using PeptideProphet and iProphet within the TPP as shown previously (Figure 1) (46; 47).” In addition, the Figure 1—figure supplement 2 is now entitled: “Combining results of three open-source database search engines in immunopeptidomics using iProphet.” Following the addition of the new table, we have decided to remove the Venn diagrams from the search identifications at 1% peptide-level FDR since this information was not essential and a bit redundant.

*4) The HLA annotation score, based on the predicted IC*_*50*_
*binding affinities from NetMHC server, is used to associate peptides with particular HLA alleles. In a similar server, Immune Epitope Database (IEDB), sometimes the values of experimentally measured and predicted values of IC*_*50*_
*can differ by a factor of 10 or more. Using the NetMHC server are the results more robust, thus allowing use of cutoff of value of 3 for the HLA annotation score*?

The results obtained using the NetMHC server are not more robust. In fact, the IEDB uses NetMHC predictors, as well as other methods. Performance, reliability and comparability of IEDB and NetMHC predictors were evaluated in details ([49]; [24]; Peters et al. 2006). Here, a reasonable cutoff value of 3 was selected mainly because >90% of the identified peptides with an annotation score above 3 have a predicted IC_50_ below 1000nM – we added this statement in the new version of the manuscript (see Materials and methods, subsection “HLA allele annotation”; and Results and Discussion). Increasing the cutoff value to 5, 50, or 500 would increase the stringency of the annotation process but would reduce significantly the number of peptides in each HLA allele-specific spectral library. In our opinion, this annotation strategy is still in a very early stage and we plan to improve it in the future. In this regard, we now clearly mention in [Supplementary-material SD8-data] (2. Annotation and Visualization): “The next version of the software tools will integrate statistical bootstrapping analysis to determine the optimal annotation cutoff value for individual datasets. The next upgrade will also integrate additional epitope prediction algorithms from IEDB and will annotate supertype peptides. We also plan to integrate a predictor-independent strategy [e.g. alignment- and clustering-based approach similar to GibbsCluster and NNalign (Andreatta et al. 2013; Nielson et al. 2009)] since the HLA annotation score used in this manuscript can only be calculated for well-characterized HLA alleles.”

Minor points:

*1) In the first paragraph of Results and Discussion, the comment* “*no reference computational framework is currently available to facilitate the analysis of such datasets" is not entirely correct, since software tools, such as MaxQuant, Perseus, or X-PRESIDENT can handle HLA-peptidome data without effort (for example: see the second reference cited,*
[2]*)*.

We agree with the reviewers. We now state (Results and Discussion, first paragraph): “Large-scale DDA-based identification of immunoaffinity purified HLA class I peptides is supported by several software tools (e.g. MaxQuant, Perseus or X-PRESIDENT) and results in thousands of unclassified peptides of various lengths. Since large HLA peptidomic datasets are generated at an increasing pace, additional computational frameworks facilitating the HLA annotation and storage of such datasets need to be developed.”

*2) In the second paragraph of Results and Discussion:* “*of all identified peptides to their respective HLA allele" is an overstatement, since significant parts of the identified peptides are not annotated to their respective HLA allele*.

We agree with the reviewers that ‘all identified peptides’ is an overstatement. We now state: ‘the majority of the identified peptides’.

*3) In the third paragraph of Results and Discussion:* “*Three synthetic EBV-derived peptides were also used to build the HLA-A02 and -B07 library. How can three peptide by useful for building two libraries? Some clarification is required*.

We agree with the reviewers that some clarification is needed. The idea here was to provide a proof-of-principle that building high-quality assay libraries from synthetic peptides could be useful for the identification of non-self HLA-bound peptides. In the revised version of the manuscript, we now mention “Synthetic EBV-derived peptides known to bind HLA-A02 or -B07 were also used to build the libraries”. We also state: “Notably, assays generated from the synthetic EBV-related class I peptides enabled the identification of one EBV-derived HLA-A02 peptide (Figure 3), thereby demonstrating that building high-quality assay libraries from synthetic class I peptides of pathogen origin could be useful for the identification of non-self HLA-bound peptides by SWATH-MS.”

*4) In the fourth paragraph of Results and Discussion:* “*Class I peptide precursors fall within the range of 400-700 Th" – this is not correct, since many peptides fall outside this range. We suggest clarifying the percentage of the peptides which fall within this range, and indicating if the loss of these peptides compensate for the additional 17% gained by use of this narrow range. Also, why was the 400-650 mass range selected*?

We thank the reviewers for this suggestion. To select the 400-650 mass range, we initially used 3,079 manually validated HLA peptide ligands isolated from 15 renal cell carcinomas (RCCs) and we define the range containing 99% of the ligands (see Figure 5). Following the publication by Mommen et al. 2014, we re-evaluated the window by opening it up to 350-900m/z and we observed a decrease in unique peptides by ∼25%. The 400-650 mass range is therefore optimal in our hands using an Orbitrap-XL but might be influenced by the dynamic range and scan speeds of individual instruments and sample concentration. Using the Triple-TOF 5600, we observed that 98% of all class I peptide precursors fall within the 400-700 mass range. Thus, we now mention in the revised version of the manuscript: “Most class I peptide precursors (∼98%) fall within the range of 400-700 Th and were divided in 30 SWATH windows of 10 Da width each.”

Author response image 2.Selection of the 400-650 mass range. 3,079 manually validated HLA class I ligands from 15 renal cell carcinomas (RCCs) were used to define the mass range containing 99% of the ligands.**DOI:**
http://dx.doi.org/10.7554/eLife.07661.034

*5) In the subsection headed “Isolation of HLA peptides”: We suggest adding the reference by*
[20]
*to the references for the method of affinity purification and LC-MS-MS analysis of MHC peptidome*.

We added the reference (to the subsection “Isolation of HLA peptides”).

*6) In the subsection “Generation of HLA allele-specific peptide spectral and assay libraries”: Please clarify that the parameters are used for Spectrast, and explain what they mean for readers that do not use Spectrast*.

We clarified that the parameters are used for Spectrast and we now guide the reader to a link where individual parameters are accurately defined. We now state in the revised version of the manuscript: “This section was adapted from [43]. The parameters below were used for Spectrast (30). Exact meaning of each parameter can be found in the following link: http://tools.proteomecenter.org/wiki/index.php?title=Software:SpectraST.”